# Novel Functional Soft Magnetic CoFe_2_O_4_/Fe Composites: Preparation, Characterization, and Low Core Loss

**DOI:** 10.3390/ma16103665

**Published:** 2023-05-11

**Authors:** Shigeng Li, Xianzhong Wang, Fangping Ouyang, Rutie Liu, Xiang Xiong

**Affiliations:** 1School of Physics and Electronics, Central South University, Changsha 410012, China; 2Department of Material and Chemistry Engineering, Pingxiang University, Pingxiang 337055, China; fmud1972@163.com; 3Jiangxi Xinke Environmental Protection Co., Ltd., Pingxiang 337000, China; 4State Key Laboratory of Powder Metallurgy, Central South University, Changsha 410083, China; llrrtt@csu.edu.cn (R.L.); xiongx@csu.edu.cn (X.X.)

**Keywords:** soft magnetic composites, in situ oxidation, nano-CoFe_2_O_4_ layer, structural characterization, low core loss

## Abstract

In this study, composites CoFe_2_O_4_/Fe were successfully synthesized by in situ oxidation, and their composition, structure, and magnetic properties have been investigated. According to the analysis of X-ray photoelectron spectrometry measure results, the cobalt ferrite insulating layer was completely coated on the surface of Fe powder particles. The evolution of the insulating layer during the annealing process has been discussed, which is correlated to effects on the magnetic properties of the composites CoFe_2_O_4_/Fe. The amplitude permeability of the composites reached a maximum of 110, and their frequency stability reached 170 kHz with a relatively low core loss of 253.6 W/kg. Therefore, the composites CoFe_2_O_4_/Fe has potential application in the field of integrated inductance and high-frequency motor, which is conducive to energy conservation and carbon reduction.

## 1. Introduction

Nowadays, the energy within a magnetic field is regulated by a magnetic core used in its electronic circuit in many modern devices (such as chargers, mobile phones, computers, electric vehicle powertrain systems, and so on). However, part of the energy is not used for work due to the loss of magnetic core, which is the main reason for the low efficiency of motors, transformers, and inductors [1]. In 2021, with the global sales of new pure electric vehicles of over 4.5 million and a staggering 340 million personal computers, even a minor reduction in the core loss, summed globally, would represent the huge energy conservation and carbon reduction of the whole society [2].

The Fe-based soft magnetic composites have received widespread concern due to their excellent magnetic properties, including high permeability, high electrical resistivity, and low core loss [3,4,5,6,7,8]. Among these properties, low core loss in a magnetic field is crucial for energy conservation and carbon reduction. Generally, the core loss includes eddy current loss and hysteresis loss. At high frequency, eddy current loss is the main factor causing core loss. Insulation coating on the particle surface to increase resistivity is the most direct and effective method to reduce the eddy current loss.

In recent years, more kinds of inorganic materials with insulative properties and ferromagnetic materials have been utilized as the coating layer to increase electrical resistivity because of their high thermal stability and resistivity. Inorganic materials, for example ZrO_2_, AlN, Al_2_O_3_, Cr_2_O_3_, TiO_2_, SiO_2_, MgO, and so on [9,10,11,12,13,14,15,16] and phosphates [17], are non-ferromagnetic, which decreases the saturation magnetization and the permeability of soft magnetic composites. Nevertheless, ferrite as an important ferromagnetic material has excellent performance, which is considered as an ideal insulating coating used in soft magnetic composites because of their high electric resistivity and low eddy current loss. Zhou et al. [18] prepared Mn-Zn ferrite (Mn_0.7_Zn_0.3_Fe_2_O_4_) nanoparticles via the chemical coprecipitation method, and the epoxy resin was developed to insulate Fe-6.5wt%Si powder. Unfortunately, the Mn-Zn and Ni-Zn ferrite nanoparticles in insulative layers could not be made a continuous layer around the powder surfaces, resulting in an increase of core loss. Therefore, the most critical factor to reduce the core loss is to prepare a uniform coating of the surface insulating layer in magnetic composites. As an advanced functional material, CoFe_2_O_4_ has been extensively investigated in multiple fields. Cao et al. [19] utilized CoFe_2_O_4_ to functionalized graphene sheets, enhancing the property of water pollution treatment. Gao et al. [20] adopted a “in-situ substitution” surface strategy to improve the catalytic activity of the CoFe_2_O_4_ by appropriately increasing the Co/Fe ratio. Meng et al. [21] prepared exchange coupled CoFe_2_O_4_/CoFe composites via in-situ hydrothermal reduction method, which enhanced the microwave absorption performance. In addition, CoFe_2_O_4_ has also been reported as a typical magnetic material. Wang et al. [22] synthesized three kinds of spinel ferrite nanocrystals MFe_2_O_4_ (M = Co, Ni, and Mn) by hydrothermal method. Magnetic measurements showed that all the samples obtained showed high saturation magnetization. The results indicate that the hydrothermal reaction time has a significant impact on the microstructure, morphology, and magnetic properties of the synthesized ferrite nanocrystals.

In this work, we have successfully prepared a uniform nano-CoFe_2_O_4_ layer coated on the surface of Fe powder though an in situ oxidation method, where the coating was directly generated by a chemical reaction with the Fe in the magnetic powder. Among spinel ferrites, CoFe_2_O_4_ is a ferromagnetic material with excellent mechanical, chemical, and thermal stability. However, to our knowledge, there are no research reports on using CoFe_2_O_4_ as the insulating layer of iron based soft magnetic composites. The evolution of the insulating layer influences the soft magnetic properties during annealing was investigated in detail. In addition, the obtained composites exhibited a low core loss of 253.6 W/kg with a large amplitude permeability, which is beneficial to energy conservation and carbon reduction in the field of high-frequency magnetic field application.

## 2. Materials and Methods

To obtain soft magnetic CoFe_2_O_4_/Fe composites, the reaction mechanism and the preparation process as shown in Figure 1 [23]. An in situ oxidation method was utilized to create a nano-CoFe_2_O_4_ insulative layer which was deposited on the surface of Fe powders. The Fe powders with an average diameter of 57 μm were synthesized via water atomization. CoCl_2_·6H_2_O (2.38 g and 0.01 mol) were mixed with 300 mL ammonia concentrations of 1.5 wt%, 2.5 wt%, and 3.5 wt% respectively, stir evenly and transfer the solutions to the reactor, then 100 g Fe powders were added to the above solution. The mixture was placed in a 500 mL autoclave and heated at 180 °C for 1 h to complete the oxidation reaction. After the oxidation reaction in situ, the obtained powder was washed with deionized water and ethanol three times before mixing with an epoxy-modified silicone resin (ESR) (2 wt%) dissolved in dimethyl benzene. The mixture was stirred with 300 r/min at 90 °C until complete evaporation of the solution. The toroidal powder cores with an outer diameter of 26.11 mm, an inner diameter of 18.00 mm, and a thickness of 4.80 mm were fabricated with cold pressure of 800 MPa. Subsequently, the cores pressed well were annealed for 1 h under an argon condition at the temperature in range of 450–600 °C.

The X-ray diffraction (XRD) patterns were measured in the 2θ rang of 10–90° at a scan speed of 0.1°/s with Advance D8 using Cu K_α_ radiation. The morphology and corresponding elemental analyses of samples after mounting were characterized by scanning electron microscopy (SEM) equipped with energy-dispersive X-ray spectroscopy (EDS; Quanta 250 FEG, Boynton Beach, FL, USA). The surface morphology of the CoFe_2_O_4_/Fe composites particles was recorded on a Lorentz-transmission electron microscopy (TEM; JEM-2100F, Akishima, Japan). XPS analysis was characterized by an X-ray photoelectron spectrometry (XPS; ESCALAB250Xi), using Al K_α_ as x-ray source. The chemical structure analyzed by Fourier transform infrared (FTIR) was collected on Bruker VERTEX 70 FTIR spectroscopy (Billerica, MA, USA) with a frequency in the range of 400 to 4000 cm^−1^, and a sample to KBr ratio of 1:100. Raman studies with a 532 nm laser were measured from 100 to 1000 cm^−1^ using a Jobin Yvon France LABRAM Aramis Raman microscopy system. The thermogravimetry and differential scanning calorimetry (TG-DSC) was performed by a thermal analysis instrument of NETZSCH STA 449C (Selb, Germany), in which the samples were heated from room temperature to 700 °C under an Ar atmosphere in alumina crucibles at a rate of 10 °C/min. Magnetic hysteresis of powder sample measured at room temperature was achieved in a vibrating sample magnetometer (VSM; Lake-shore 7400-s) in applied fields of −20,000 and 20,000 Oe at 25 °C. The amplitude permeability and core loss of the CoFe_2_O_4_/Fe composites were performed on an auto-testing system for magnetic materials (SY8258B-H/m) with a maximum applied induction of 50 mT within the frequency range of 10–170 kHz.

## 3. Results and Discussion

Figure 2 shows the XRD data with sample oxidized at different contents of NH_3_·H_2_O (1.5, 2.5, and 3.5 wt%). According to JCPDS Card No. 87-0721, the diffraction peaks at 2θ values of ca. 44.70°, 64.98°, and 82.30° were from the (110), (200) and (211) planes of the α-Fe phase. And unexpectedly, only peaks originating from the raw powders were detected without the observation of any peaks corresponding to the CoFe_2_O_4_ phase of the oxidized powders. The reason for this result may be that the amount of CoFe_2_O_4_ phase is too low to be detected by XRD.

Figure 3 shows the SEM micrograph with EDS results of the oxidized Fe powders. It can be seen that the surface of oxidized iron powder particles in the morphology under SEM is rough, indicating that there is an oxidation layer on the surface. The element of cobalt, iron, and oxygen from corresponding area could be seen obviously. This suggests that a CoFe_2_O_4_ layer was formed. Regarding the depth of analysis, a comparison of the strengths of the oxygen and iron peaks indicates that the content of CoFe_2_O_4_ is relatively low.

The corresponding EDS mapping of the oxidation layer was further used to confirm the chemical composition. Figure 4 shows the SEM image and the corresponding EDS spatial elemental mapping of Co, Fe, and O for the Fe powder coated with the insulating oxide layer. The EDS profile ulteriorly proved that Co, Fe, and O were uniformly deposited on the surface of the Fe powder particle, which is critical to ensure a relatively low eddy current loss of the soft magnetic composites.

To observe the thickness of the CoFe_2_O_4_ layer, the Lorentz transmission electron microscope morphology of oxidized powder particles was analyzed, as shown in Figure 5. The external grey layer was coated on the internal black part. The CoFe_2_O_4_ insulating layer was deposited on the surface of Fe particles with a thickness of approximately 45 nm.

XPS was measured to determine the oxidation states of the insulating layer on the oxidized powders. The XPS patterns of the oxidized Fe powders sputtered for 1000 s by Ar^+^ cation can be seen in Figure 6. High-resolution XPS spectra of the 2p peaks of Co and Fe are described in Figure 6a,b. The electron binding energy of the Co 2p level was measured: the Co 2p1/2 and 2p3/2 peaks were located at approximately 795.7 eV and 780.2 eV, respectively, along with characteristic shake-up satellite peaks identified at approximately 802.1 eV, and 785.5 eV, confirming the existence of Co^2+^ species [24]. The XPS Fe 2p spectrum exhibited two peaks corresponding to Fe 2p1/2 and Fe 2p3/2 located at approximately 724.4 eV and 711.1 eV, respectively, with a typical satellite peak at approximately 719.4 eV, which were assigned to Fe in the 3+ oxidation state [25]. The survey spectrum of oxidized powder is shown in Figure 6c, which indicate the existence of cobalt, iron, oxygen, and carbon. The analysis results of the XPS measurements further demonstrated that the insulating layer comprised with CoFe_2_O_4_.

A weak peak at 706.6 eV was observed for the sample oxidized by 1.5 wt% NH_3_·H_2_O, which corresponds to the Fe 2p3/2 binding energy in metallic Fe, indicating an incomplete CoFe_2_O_4_ insulating layer as shown in Figure 7. During the in situ oxidation process, the nano-CoFe_2_O_4_ layer will grow on the surface of the bare Fe. Figure 7d shows that a weak satellite peak at 719.4 eV was observed in the XPS spectrum, which corresponds to Fe^3+^. It is noteworthy that the intensity of satellite peak increases with the increase of reaction concentration.

The chemical functional groups of the CoFe_2_O_4_/Fe composites were performed by FT-IR spectroscopy in the range 400–4000 cm^−1^, as shown in Figure 8. The broad absorption bands were observed at 3437, 1637, 1379 and 1058 cm^−1^ due to the -OH, NH_4_^+^ and NH_3_ vibration, whose peaks are considered to be appeared due to H_2_O in the air and the adsorbed NH_3_ on the CoFe_2_O_4_/Fe composites [26,27]. From the spectrum, the absorption bands at 779 and 465 cm^−1^ were assigned to Fe-O stretching. Furthermore, the absorption peak at 572 cm^−1^ corresponded to the vibration of Co-O [28]. The above results further demonstrated the nano-CoFe_2_O_4_ layer was formed on the on the surface of the Fe powder.

Raman spectroscopy is an important characterization technique for investigating the vibrational motion of molecules. The Raman spectroscopy of the oxidized powder sample was shown in Figure 9. The characteristic peaks in the sample appeared as sharp peaks at 213 and 275 cm^−1^ and shoulder peaks appeared at 390, 481, 590, and 654 cm^−1^, which confirmed the formation of a CoFe_2_O_4_ cubic spinel ferrite [22]. Since the Raman peaks at approximately 590 cm^−1^ usually correspond to the nature of the tetrahedra in the spinel, the low-frequency mode of approximately 490 cm^−1^ is related to an octahedral structure [22].

According to the in-situ oxidation reaction mechanism and above measurement results, with the increase of ammonia concentration, OH^−^ and NH^4+^ in the solution increase, which makes the insulating coating thicker. Therefore, the reaction concentration will control the thickness of the insulating layer and thus affects the magnetic properties of the CoFe_2_O_4_/Fe composites.

Figure 10 provides the variations of magnetization as a function of an applied magnetic field, as found by measuring samples of the Fe powders and those undergoing different reaction concentrations. The magnetic moment as a function of the magnetic field was measured over a range from −20,000 to 20,000 Oe at 25 °C. The values of saturated magnetization (*M*_s_) were decreased sequentially from 207 to 203 emu/g with increasing ammonia concentration, because of the insulating CoFe_2_O_4_ layer with lower *M*_s_ [29]. The *M*_s_ values of the oxidized Fe powder are consistent with that of the raw powder (210 emu/g). This is attributed to the formation of the ferromagnetic CoFe_2_O_4_ coating by the in situ oxidation method for the prevention of magnetic dilution.

It is necessary to research the thermal stability of CoFe_2_O_4_/Fe composites to determine the annealing temperature. The TG-DSC curves were measured for the CoFe_2_O_4_/Fe composites powders on the NETZSCH STA 449C, as shown in Figure 11. It can be clearly seen that the mass was decreased at approximately 190 °C due to thermal degradation of the epoxy-modified silicone resin layer, which decomposed slowly with increasing temperature. The DSC curve exhibits an endothermic peak at 562 °C, which indicates that the CoFe_2_O_4_ began to decompose, leaving Fe_2_O_3_ [30].

In order to explore the surface phase transformation of CoFe_2_O_4_/Fe composites during annealing, the SEM micrographs and the element distribution of the polished surface of samples annealed in Ar atmosphere at various temperatures from 500 °C to 600 °C for 1 h were shown in Figure 12. After annealing at 500 °C, the surface of CoFe_2_O_4_/Fe composites was still covered with a complete and uniform insulating layer. However, the thickness of the insulating layer decreased or even disappeared with increasing annealing temperature. The insulating layer (dark part) of the CoFe_2_O_4_/Fe composites contained Fe, O, Co, and Si. The coating layer contained Si because ESR partially decomposed into SiO_2_. It is noteworthy that the comparison of the Si intensities at different annealing temperatures indicated that the ESR layer gradually decomposed with increasing temperature, which is consistent with the DSC analysis.

To study the evolution of the CoFe_2_O_4_/Fe composites during annealing, Figure 13 shows the XPS spectra taken from the samples after annealing at 500 °C, 550 °C and 600 °C for 1 h. The Fe 2p1/2 and Fe 2p3/2 peaks are located near 724.4 eV and 710.1 eV, respectively, with a more obvious satellite peak near 719.8 eV, indicating that the CoFe_2_O_4_ is exposed on the surface, which proves that the silicone resin coating of the samples annealed at 500 °C was partially decomposed. In addition, As shown in Figure 13b, a weak peak was surveyed near 706.5 eV, which corresponds to the Fe 2p3/2 peak of the metallic Fe, indicating that part of CoFe_2_O_4_ is reduced to Fe during annealing [25]. Figure 13c shows the samples annealed at 550 °C, the positions of Fe 2p1/2 peak and Fe 2p3/2 peak remain unchanged, but the satellite peak intensity corresponding to Fe^3+^ near 719.8 eV is significantly higher than the test results at 500 °C, indicating that the silicone resin is further decomposed and more CoFe_2_O_4_ is exposed to the particle surface. Moreover, the peak corresponding to metallic Fe near 706.5 eV is more obvious than that at 500 °C, illustrating that the amount of reduced Fe is further increased. For the samples annealed at 600 °C as shown in Figure 13d, the satellite peak of Fe^2+^ appeared near 715.7 eV indicating that Fe^3+^ was reduced to Fe^2+^. However, the peak intensity of metallic Fe at 706.5 eV is lower than that at 550 °C, indicating that the reduced Fe was oxidized. Therefore, it can be inferred that CoFe_2_O_4_ was reduced to FeO by Fe when the sample was annealed at 600 °C.

On the basis of DSC, SEM, EDS, and XPS test results, the evolution of the insulating layer during annealing can be inferred, and its schematic diagram is shown in Figure 14. With the annealing temperature reaching 200 °C, the silicone resin begins to decompose to form carbon, carbonyl, or other reducing groups on the particle surface, as shown in Figure 14b. When the temperature is further increased, the reducing group begins to react with CoFe_2_O_4_ to form Fe (Figure 14c). According to the XPS analysis results, the reaction temperature starts before 500 °C, and the reaction mainly occurs at the interface between CoFe_2_O_4_ and silicone resin. Annealing at a further raised temperature to 562 °C, the reduced Fe and Fe inside the particles will react with CoFe_2_O_4_ to form FeO. The reaction mainly occurs at the interface between Fe and CoFe_2_O_4_ and the interface between CoFe_2_O_4_ and silicone resin, as shown in Figure 14d.

Figure 15 shows the change in amplitude permeability (*μ*_a_) of measurement frequency for CoFe_2_O_4_/Fe composites. The *μ*_a_ of CoFe_2_O_4_/Fe composites annealed at 450 and 500 °C exhibited good frequency stability in the range of 10 kHz to 170 kHz. However, when the annealing temperature was 550 °C, the *μ*_a_ of CoFe_2_O_4_/Fe composites was decreased slightly with increasing frequency, mainly due to degradation of the organic insulating layer. In particular, when the annealing temperature reached 600 °C, the *μ*_a_ decreased sharply with increasing frequency. This phenomenon can be explained by the influence of the organic and inorganic insulating layers. On the one hand, the further decomposition of the organic layer made the insulating layer incomplete. On the other hand, it is known that the inorganic nano-CoFe_2_O_4_ layer decomposed at elevated annealing temperatures, resulting in an incomplete CoFe_2_O_4_ insulation coating. In addition, the *μ*_a_ of CoFe_2_O_4_/Fe composites reached a maximum value of approximately 110 annealed at 500 °C.

Figure 16 shows the core loss (*P_s_*) versus the frequency of CoFe_2_O_4_/Fe composites, which mainly composed of the hysteresis loss *P_h_* and eddy current loss *P_e_*, given by the following equation [31,32]:(1)$$P_{s}=P_{h}+P_{e}=f\oint HdB+\frac{CB^{2}f^{2}d^{2}}{\rho }$$
where “*H*” refers strength of the magnetic field, “*B*” refers the flux density, “*f*” refers the frequency, “*d*” refers the thickness, “*C*” refers the constant of proportionality, and “*ρ*” refers the resistivity.

The *P_s_* of the CoFe_2_O_4_/Fe composites samples slightly decreased with annealing temperatures ranging from 450 to 500 °C. This mainly resulted from the reduced residual stresses and decreased coercivity caused by the annealing treatment, decreasing the hysteresis loss of *P*_s_. The *P*_s_ was increased with the increasing temperature range from 500 to 550 °C, which was mainly caused by the further decomposition of organic coating leading to an increase in the eddy current loss. However, the *P*_s_ of the sample annealed at 600 °C sharply increased by 271% compared with the sample annealed at 500 °C. Because the nano-CoFe_2_O_4_ insulating layer would be damaged and could not completely coat the surface of Fe powders. The incomplete insulating layer can lead to increased eddy current loss in the CoFe_2_O_4_/Fe composites. Moreover, the core loss of the samples oxidized by different ammonia concentrations annealed at 500 °C, as shown in Figure 16d. The core loss exhibited evident decreasing trend depending on increasing ammonia concentration, resulting in a minimum value of 253.6 W/kg recorded at 100 kHz with a mass percentage of 2.5%. At the ammonia concentration of 1.5 wt%, the incomplete CoFe_2_O_4_ insulative layer was unable to provide a significant electrical resistivity for the CoFe_2_O_4_/Fe composites, giving rise to a large core loss. However, the ammonia concentration was increased to a content of 3.5 wt%, causing the growth of CoFe_2_O_4_. This would lead to a large hysteresis loss due to a large coercive force in the CoFe_2_O_4_ rather than in the Fe.

From what has been mentioned above, it can be found that the oxidized CoFe_2_O_4_ insulating layer on the iron powder surface in situ had excellent thermal stability and insulation properties, which can significantly reduce the eddy current loss of the CoFe_2_O_4_/Fe composites samples. Combined with the annealing temperature and ammonia concentration, 500 °C and 2.5 wt% were the optimal processes for obtaining good soft magnetic properties with a high amplitude permeability (approximately 110) and low core loss (253.6 W/kg, 50 mT 100 kHz) in the CoFe_2_O_4_/Fe composites.

## 4. Conclusions

In summary, CoFe_2_O_4_/Fe composites consisting of Fe powders coated with nano-CoFe_2_O_4_ insulating layers were produced by in situ oxidation and mixing with the ESR. It is proved through analyses of the test data by various instruments that the Fe powders were coated with a nano-CoFe_2_O_4_ insulating layer. The decomposition of CoFe_2_O_4_ to Fe_2_O_3_ occurred at an elevated annealing temperature of 562 °C from the DSC result. The magnetic properties of CoFe_2_O_4_/Fe composites and the evolution of the insulating layer during annealing were investigated. With optimized annealing, the CoFe_2_O_4_/Fe composites exhibited excellent advantages in amplitude permeability and core loss. The overall results of this study indicate that in situ oxidation and annealing treatment can be used to fabricate excellent amplitude permeability and low core loss Fe-based soft magnetic composites, which are conducive to energy conservation and carbon reduction.

## Figures and Tables

**Figure 1 materials-16-03665-f001:**
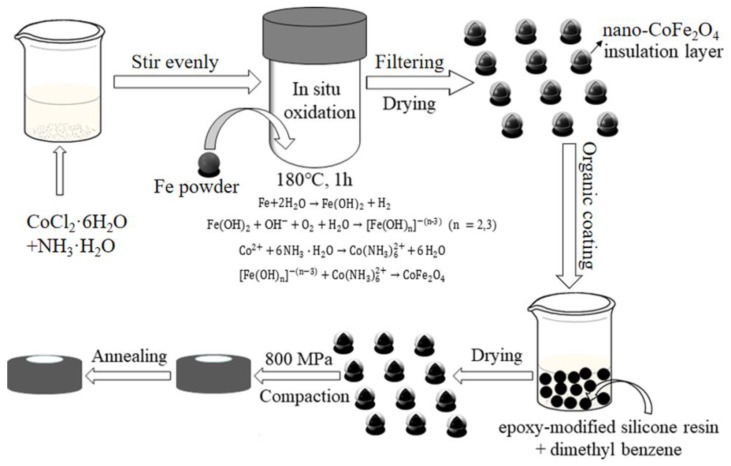
Schematic diagram of preparation process of soft magnetic CoFe_2_O_4_/Fe composites.

**Figure 2 materials-16-03665-f002:**
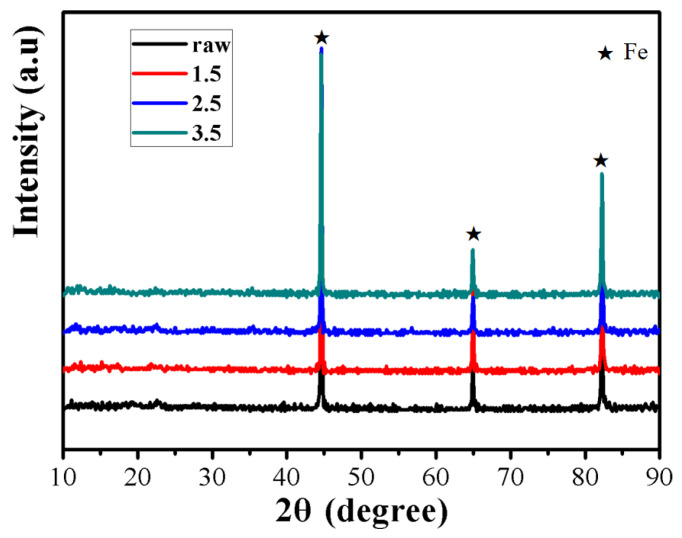
XRD patterns under different reaction concentrations.

**Figure 3 materials-16-03665-f003:**
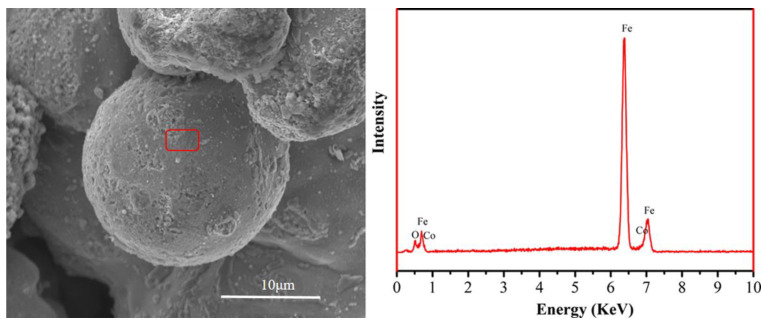
SEM micrograph and EDS spectrum of oxidized Fe powder.

**Figure 4 materials-16-03665-f004:**
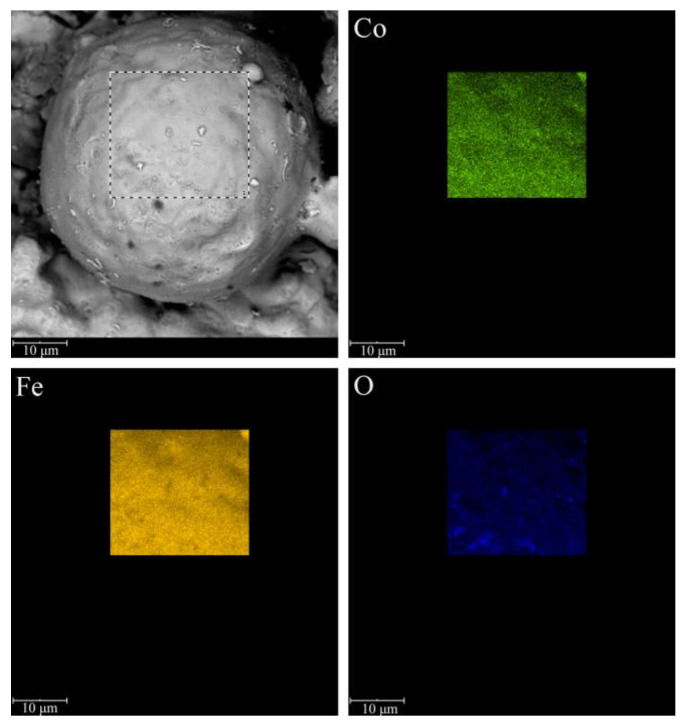
SEM micrograph and EDS elemental distribution maps of the oxidized Fe powder prepared with 2.5 wt% NH_3_·H_2_O.

**Figure 5 materials-16-03665-f005:**
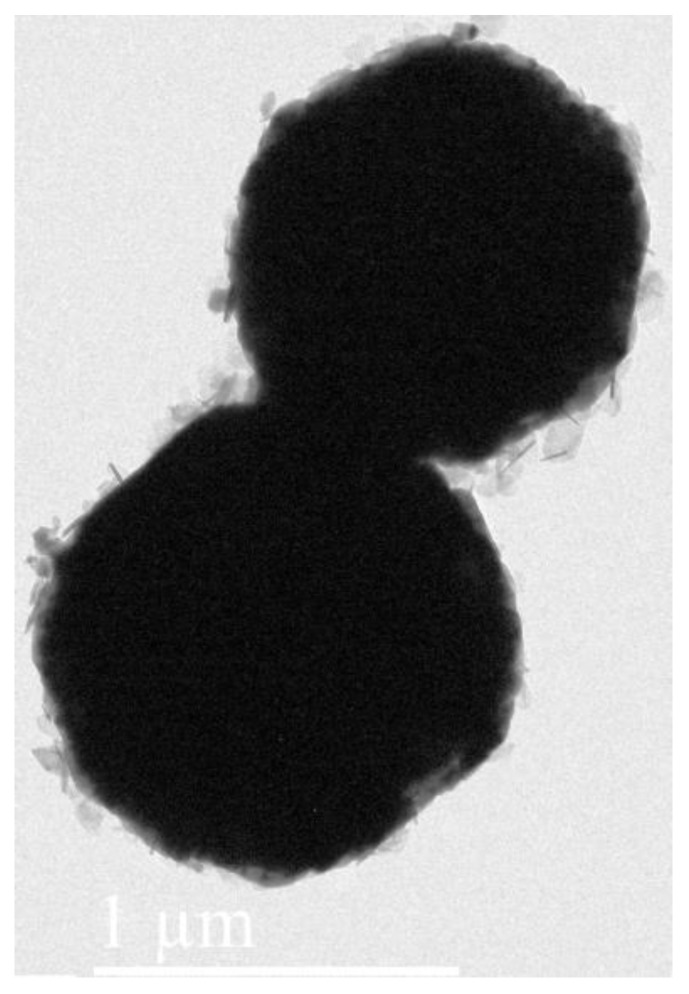
TEM image of CoFe_2_O_4_/Fe composites.

**Figure 6 materials-16-03665-f006:**
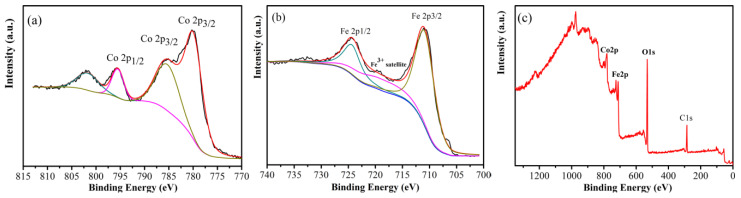
High-resolution XPS spectra of (**a**) Co 2p and (**b**) Fe 2p peaks and (**c**) a full-scan spectrum of the Fe powder oxidized with 2.5 wt% NH_3_·H_2_O.

**Figure 7 materials-16-03665-f007:**
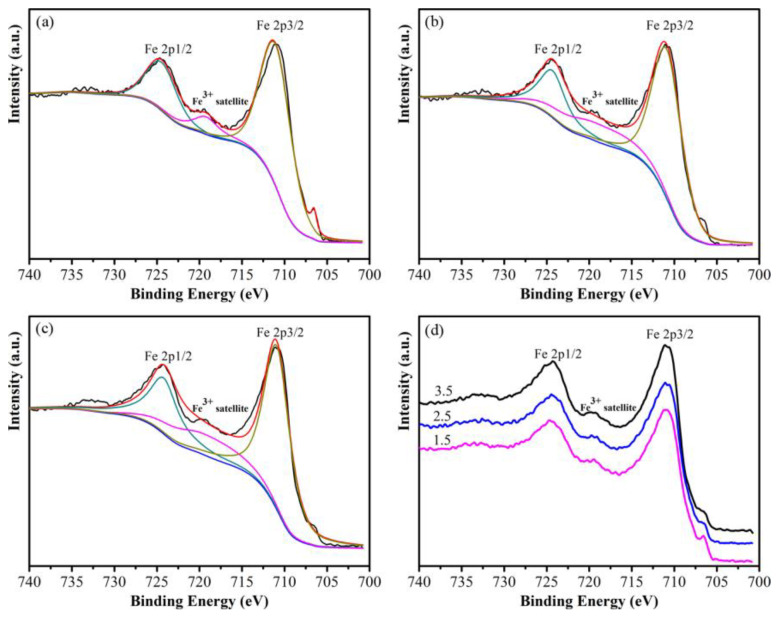
XPS spectra of the Fe 2p peaks taken from the Fe powders undergone different reaction concentrations: (**a**) 1.5 wt%, (**b**) 2.5 wt%, (**c**) 3.5 wt%, (**d**) undergone different reaction concentrations.

**Figure 8 materials-16-03665-f008:**
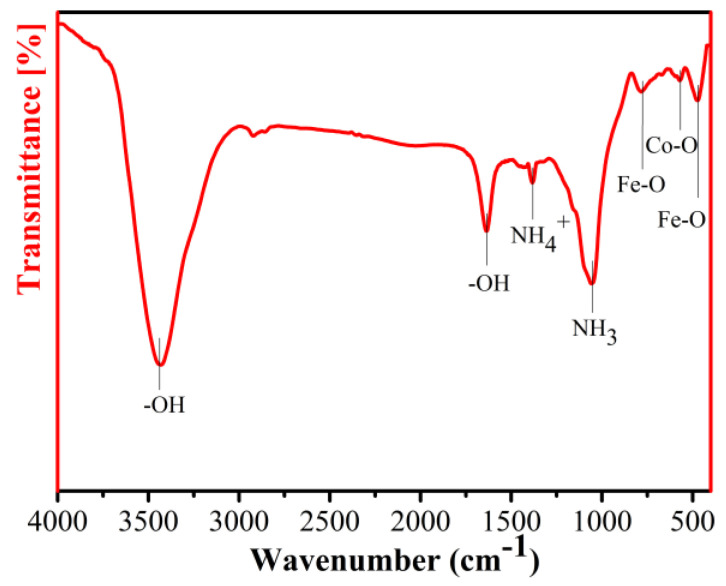
FT-IR spectrum of the CoFe_2_O_4_/Fe composites.

**Figure 9 materials-16-03665-f009:**
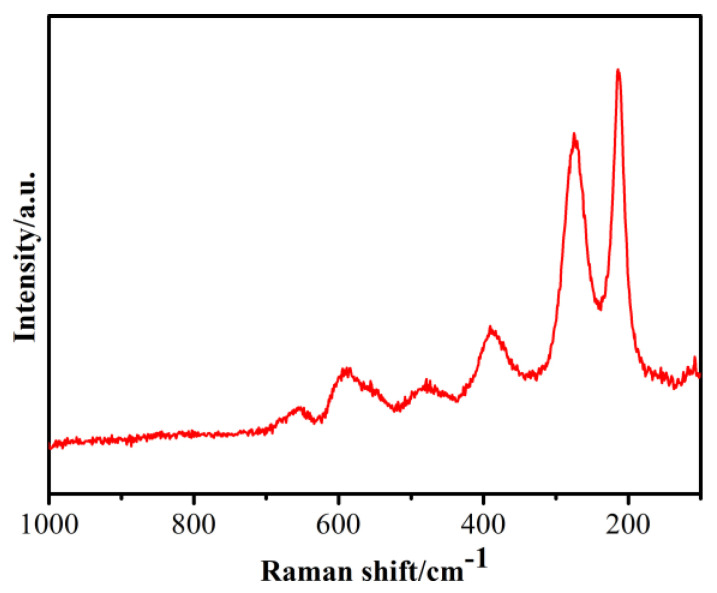
Raman spectroscopy result for the oxidized Fe powder.

**Figure 10 materials-16-03665-f010:**
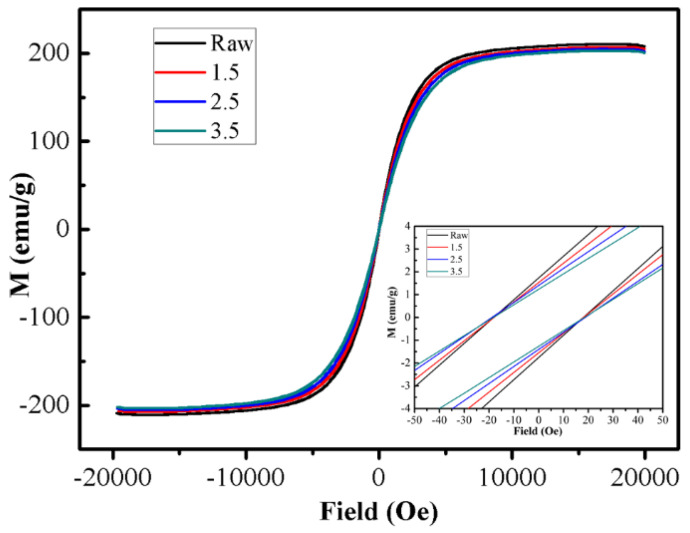
VSM spectra at 25 °C of Fe and composite powders coated with nano-CoFe_2_O_4_ layer.

**Figure 11 materials-16-03665-f011:**
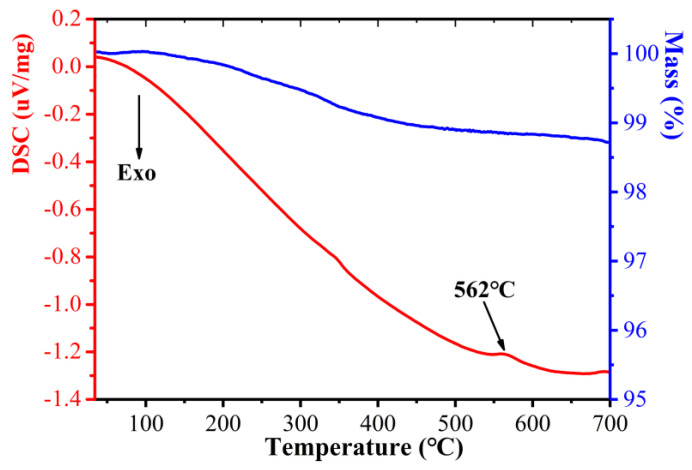
TG-DSC curve of the CoFe_2_O_4_/Fe composites.

**Figure 12 materials-16-03665-f012:**
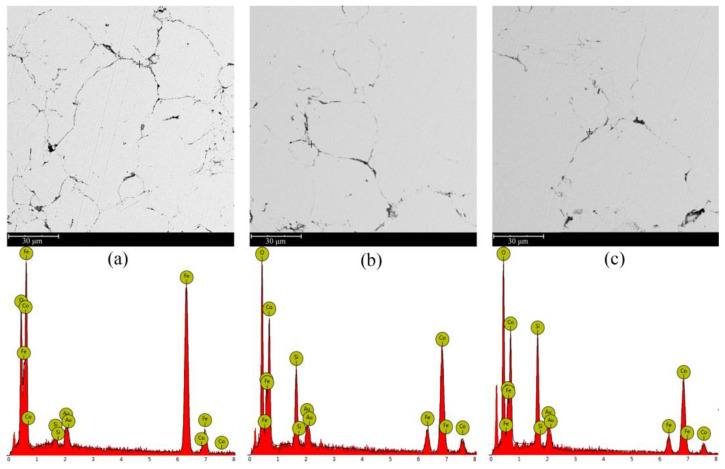
EDS diagram of the sample after annealing at (**a**) 500 °C, (**b**) 550 °C and (**c**) 600 °C.

**Figure 13 materials-16-03665-f013:**
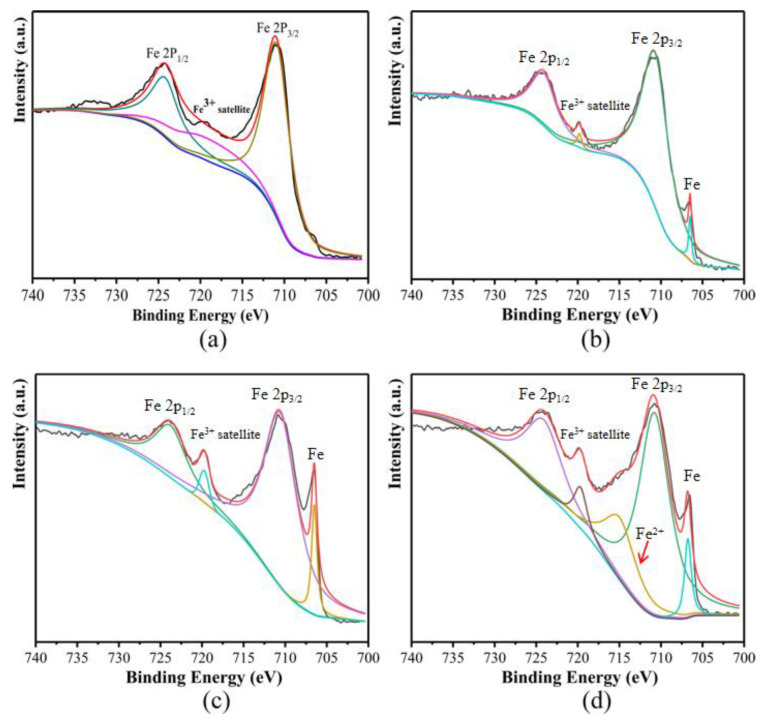
XPS spectra of the Fe 2p peaks taken from the (**a**) CoFe_2_O_4_/Fe composites and (**b**) those annealed at 500 °C, (**c**) 550 °C, and (**d**) 600 °C.

**Figure 14 materials-16-03665-f014:**
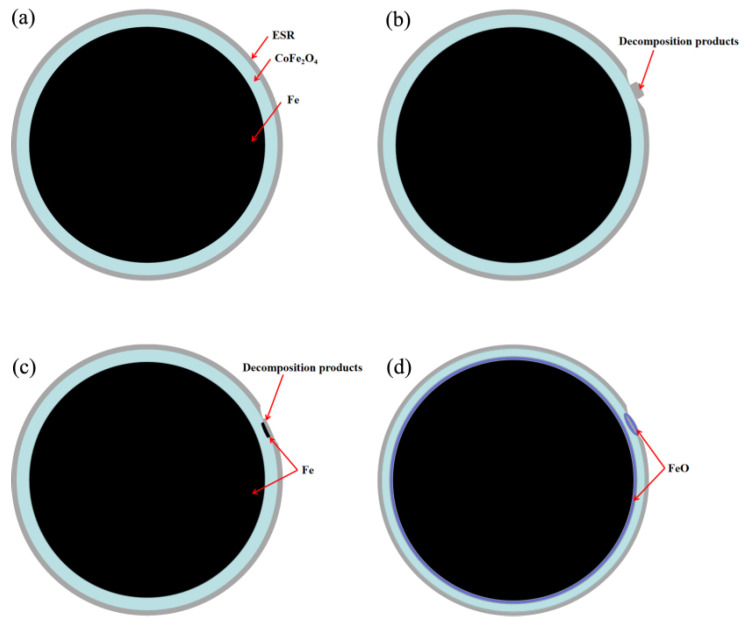
Schematic drawings illustrating the evolution of the CoFe_2_O_4_/ESR during annealing, showing (**a**) Fe powder coated with CoFe_2_O_4_/ESR, (**b**) ESR begins to decompose to form reductive group, (**c**) The reduced group reacts with CoFe_2_O_4_ to form Fe and (**d**) FeO formed by the reaction of CoFe_2_O_4_ with Fe.

**Figure 15 materials-16-03665-f015:**
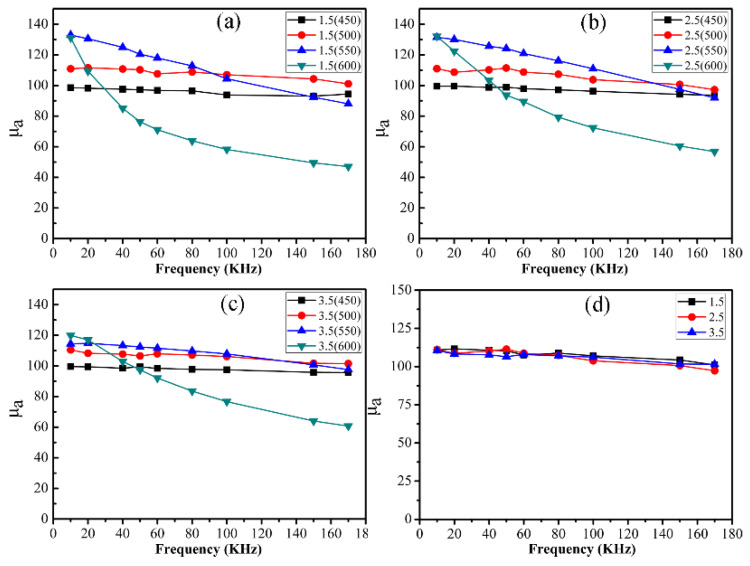
The amplitude permeability spectra for the CoFe_2_O_4_/Fe composites: (**a**) 1.5 wt%, (**b**) 2.5 wt%, (**c**) 3.5 wt%, and (**d**) annealed at 500 °C.

**Figure 16 materials-16-03665-f016:**
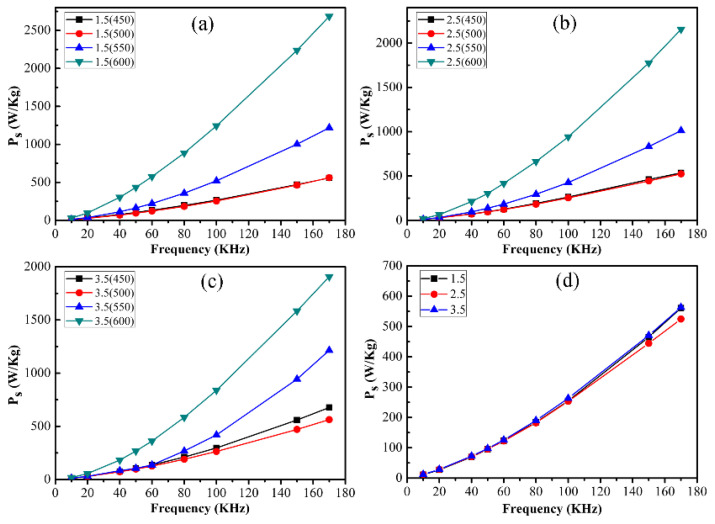
Core loss versus frequency of the CoFe_2_O_4_/Fe composites. (**a**) 1.5 wt%, (**b**) 2.5 wt%, (**c**) 3.5 wt%, and (**d**) annealed at 500 °C.

## Data Availability

Data sharing is not applicable.

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
