# Peer review of "Novel Functional Soft Magnetic CoFe2O4/Fe Composites: Preparation, Characterization, and Low Core Loss"

_materials, 2023, doi:10.3390/ma16103665_

Round 1

Reviewer 1 Report

The manuscript of Li et al. is devoted to the investigation of novel magnetic soft composites with low magnetic losses. The subject regards the important research of novel soft magnetic with very small losses that are necessary for the energetic transition involving telecommunications, energy generation transformation and mobility. Actually most of soft magnetic materials in the range of 100 kHz uses FeSi,  Metaglass or similar Fe based alloys. Here the authors propose a soft composite formed by micrometric metallic Fe particles capped by Cobalt ferrite (CFO) layer of tens of nanometers thick. The CFO layer acts an insulator layer to reduce eddy currents (as many other examples in literature, but do not considered). Authors propose that the magnetic hardness can play a role in the magnetic properties, but this is not demonstrated. The synthesis of the oxide layer with different precursors and the effect of the thermal sintering are explored. The authors investigate the structural, composition, Raman and static and dynamic magnetic properties of powders or magnets. Most characterizations regard the macroscopic material and in the case of nanolayer, its composition is only investigated. The explanation of the dynamic magnetic behavior is discussed without experimental support. This is an article mostly devoted to bulk materials and the material cannot be described as a nanocomposite. The reported values of losses and permeabilities are below of commercial ones but as an alternative is interesting.

Author Response

Dear Reviewer:

Thank you for your letter and comments concerning our manuscript entitled "Novel Functional Soft Magnetic CoFe2O4/Fe Composites: Preparation, Characterization and Low core loss" (nanomaterials-2320050). Those comments are all valuable and very helpful for revising and improving our paper, as well as the important guiding significance to our researches. The responds to the comments are as flowing:

Reviewer: 1 

The manuscript of Li et al. is devoted to the investigation of novel magnetic soft composites with low magnetic losses. The subject regards the important research of novel soft magnetic with very small losses that are necessary for the energetic transition involving telecommunications, energy generation transformation and mobility. Actually most of soft magnetic materials in the range of 100 kHz uses FeSi, Metaglass or similar Fe based alloys. Here the authors propose a soft composite formed by micrometric metallic Fe particles capped by Cobalt ferrite (CFO) layer of tens of nanometers thick. The CFO layer acts an insulator layer to reduce eddy currents (as many other examples in literature, but do not considered). Authors propose that the magnetic hardness can play a role in the magnetic properties, but this is not demonstrated. The synthesis of the oxide layer with different precursors and the effect of the thermal sintering are explored. The authors investigate the structural, composition, Raman and static and dynamic magnetic properties of powders or magnets. Most characterizations regard the macroscopic material and in the case of nanolayer, its composition is only investigated. The explanation of the dynamic magnetic behavior is discussed without experimental support. This is an article mostly devoted to bulk materials and the material cannot be described as a nanocomposite. The reported values of losses and permeabilities are below of commercial ones but as an alternative is interesting.

  1. The CFO layer acts an insulator layer to reduce eddy currents (as many other examples in literature, but do not considered).

We have added literature description on the research progress of CoFe2O4 in the introduction section.

  1. Authors propose that the magnetic hardness can play a role in the magnetic properties, but this is not demonstrated.

We have added a comparison of sample coercivity in our discussion of Figure 10.

  1. The explanation of the dynamic magnetic behavior is discussed without experimental support.

We discussed the magnetic dynamic performance based on experimental data. 

  1. This is an article mostly devoted to bulk materials and the material cannot be described as a nanocomposite.

Based on your suggestions, we have modified the nonacomposites in the article to composites. 

In addition, we have marked the revised parts of the article in red.

    We tried our best to improve the manuscript. Thank you very much for your comments and suggestions.  

Best wishes.

Sincerely

Shigeng Li, Fangping Ouyang

Professor of Materials Science

School of Physics and Electronics,

Central South University

Tel: +86-15079927360

Fax: +86-15079927360

Email: [email protected]; [email protected].

Reviewer 2 Report

The manuscript "Novel Functional Soft Magnetic CoFe2O4/Fe Nanocomposites: Preparation, Characterization and Low core loss", written by Li et al., describes an interesting approach towards a synthesis and application of functionalized, magnetic Co@Fe2O4/Fe materials. The theme is relatively interesting and obtained results could be important form many researchers working in catalysis / sensing. The work is well written and results are well described and discussed. I have few minor comments:

1. Could the authors show also SEM micrograph of the material after the interaction with epoxy modified silicon resin? Before annealing?

2. Authors should also show Raman and IRE spectra obtained during each stage of the synthesis and show / describe spectral changes.

3. Font size in the figure 1 should be increased

Author Response

Dear Reviewer:

Thank you for your letter and comments concerning our manuscript entitled "Novel Functional Soft Magnetic CoFe2O4/Fe Composites: Preparation, Characterization and Low core loss" (nanomaterials-2320050). Those comments are all valuable and very helpful for revising and improving our paper, as well as the important guiding significance to our researches. The responds to the comments are as flowing:

Reviewer: 2 

The manuscript "Novel Functional Soft Magnetic CoFe2O4/Fe Nanocomposites: Preparation, Characterization and Low core loss", written by Li et al., describes an interesting approach towards a synthesis and application of functionalized, magnetic CoFe2O4/Fe materials. The theme is relatively interesting and obtained results could be important form many researchers working in catalysis / sensing. The work is well written and results are well described and discussed. I have few minor comments:

  1. Could the authors show also SEM micrograph of the material after the interaction with epoxy modified silicon resin? Before annealing?

Due to the magnetism of the sample, it is necessary to perform sample embedding treatment during SEM testing, and the required temperature for sample embedding is 180 ℃. At this temperature, the epoxy modified silicon resin has decomposed, which is not conducive to observing its SEM morphology. The article mainly focuses on the characterization of surface inorganic insulation layers and their impact on magnetic properties. I hope to receive your recognition, Thank you very much.

  1. Authors should also show Raman and IRE spectra obtained during each stage of the synthesis and show / describe spectral changes.

The main purpose of using Raman and infrared spectroscopy to test the samples is to further determine that the surface of magnetic particles has an insulating layer. There is less emphasis on the spectral changes that occur during the preparation process. I hope to receive your recognition and support. Thank you very much for your valuable advice.

  1. Font size in the figure 1 should be increased.

    We have increased the size of figure 1.

In addition, we have marked the revised parts of the article in red.

    We tried our best to improve the manuscript. Thank you very much for your comments and suggestions.  

Best wishes.

Sincerely

Shigeng Li, Fangping Ouyang

Professor of Materials Science

School of Physics and Electronics,

Central South University

Tel: +86-15079927360

Fax: +86-15079927360

Email: [email protected]; [email protected].

Reviewer 3 Report

The manuscript “Novel Functional Soft Magnetic CoFe2O4/Fe Nanocomposites: Preparation, Characterization and Low core loss” by Shigeng Li et al. describes as well depicted in its title the synthesis and mainly the characterization of a soft magnetic nanocomposite.

In general, the manuscript is well-written. The manuscript and the topic might be suitable to the journal Nanomaterials and the readership of Nanomaterials.

However, there are also multiple smaller issues with this manuscripts which need to be overcome.

I recommend a major revision of this manuscript, since I believe that there is no need to conduct new experiments but significant parts of the manuscript need to be revised and rewritten.

In general there are many figures in this manuscript and only few references. The discussion does not compare the finding to literature.

Introduction:

The topic is nicely introduced in the beginning stating a problem which needs to be solved. However, the state of the art is described very poorly. There is close to no state of the art described concerning cobalt ferrite particles and cobalt ferrite/iron nanocomposites even though there is a lot of literature concerning this topic (even though mainly for different applications).

Thus, I recommend to thoroughly revise this introduction and especially describe the synthesis of cobalt ferrites (and the multiple approaches) in more detail in the introduction.

Materials and Methods:

This section needs to be significantly improved. The description of methods does not allow to reproduce and comprehend the manuscript.

For the synthesis, the stirring velocity is unclear. Also it is not clear what kind of iron powders (e.g. which size) have been used for the synthesis.

The characterization methods and the preparation for the characterization is also not very well described. How did you prepare your samples for SEM and TEM? In which geometry did you measure XRD (powder diffraction in transmission)?

Which X-ray source has been used for XPS?

The same is true for IR spectroscopy. How did you conduct the measurements? In transmission geometry with KBr? Please state your preparation and experiments!

The same is true for Raman spectroscopy. At which power did you conduct your measurements (since they are prone to further oxidation)?

For TGA and DSC it is not clear if the experiments have been conducted in inert gas or in air. Moreover, there is no statement for the heating rate.

Results and Discussion:

Can you discuss your XRD results with literature?

Figure 3: There is no scaling bar for the SEM image. Please add one.

In the EDS: Where exactly can one observe cobalt? I just see iron which strongly overlays cobalt.

Figure 4: The scaling bars are not readable. It is also not clear how you identified each element? Can you add how the color relates to each element? I think you related the coloring to a specific energy?

Figure 5: You show particles larger than 1 µm. How can this be called a nanocomposite?

Figure 6: How did you fit your spectra? This is not described. Also it seems that one fit for Co2p3/2 is missing.

Figure 7: Moreover, no peak corresponding to metallic Fe was observed for the sample oxidized by 2.5 wt% NH3·H2O, which suggests that a complete nano-CoFe2O4 layer was formed. How does this observation lead to this statement? You just observe iron. You observe oxides here and there might also be some carbon and therefore no elemental iron can be observed. Unless you show the cobalt oxide, you cannot make this statement.

Figure 8: Can you please reference your peak assignments? “From the spectrum, the absorption bands at 1058, 779, and 465 cm-1 were assigned to Fe-O stretching

1379 cm-1 is usually not assigned to an OH vibration. Also there is no Fe-O vibration at around 1000 cm-1. Both peaks rather indicate carbon, which is also visible in the XPS.

Figure 9: Can you turn the figure. From high to low wavenumbers?

Also can you please reference your spinel statement?

Figure 10: Please indicate the temperature in the figure caption.

Can you also compare your saturation magnetization as well as the remanence of your composites with literature?

Figure 11: There are no units on the left y-axis. Please add the units to show the correct range.

Also it is unclear why the peak at 562 should lead to oxidation. One would expect an increase in weight if oxygen is taken up. Also what happens to the cobalt? Please reference your findings with observations in literature.  Please also indicate the atmosphere which was used for your experiments.

No reference in XPS discussion

Figure 12: Legends and scaling bar are not readable. Can you increase the font for the x-axis of EDS and the font for the scaling bar (or mention the scale in the figure caption).

Author Response

Dear Reviewer:

Thank you for your letter and comments concerning our manuscript entitled "Novel Functional Soft Magnetic CoFe2O4/Fe Composites: Preparation, Characterization and Low core loss" (nanomaterials-2320050). Those comments are all valuable and very helpful for revising and improving our paper, as well as the important guiding significance to our researches. The responds to the comments are as flowing:

Reviewer: 3 

The manuscript “Novel Functional Soft Magnetic CoFe2O4/Fe Nanocomposites: Preparation, Characterization and Low core loss” by Shigeng Li et al. describes as well depicted in its title the synthesis and mainly the characterization of a soft magnetic nanocomposite. In general, the manuscript is well-written. The manuscript and the topic might be suitable to the journal Nanomaterials and the readership of Nanomaterials. However, there are also multiple smaller issues with this manuscripts which need to be overcome. I recommend a major revision of this manuscript, since I believe that there is no need to conduct new experiments but significant parts of the manuscript need to be revised and rewritten. In general there are many figures in this manuscript and only few references. The discussion does not compare the finding to literature.

Introduction:

The topic is nicely introduced in the beginning stating a problem which needs to be solved. However, the state of the art is described very poorly. There is close to no state of the art described concerning cobalt ferrite particles and cobalt ferrite/iron nanocomposites even though there is a lot of literature concerning this topic (even though mainly for different applications). Thus, I recommend to thoroughly revise this introduction and especially describe the synthesis of cobalt ferrites (and the multiple approaches) in more detail in the introduction.

We have made significant revisions to the introduction and added a literature review on the research progress of CoFe2O4, along with corresponding references.

Materials and Methods:

This section needs to be significantly improved. The description of methods does not allow to reproduce and comprehend the manuscript.

We have made corresponding modifications to the Materials and Methods section.

For the synthesis, the stirring velocity is unclear. Also it is not clear what kind of iron powders (e.g. which size) have been used for the synthesis.

    We have made corresponding modifications in the manuscript. The Fe powders with an average diameter of 57 μm were synthesized via water atomization. The mixture was stirred with 300 r/min at 90 ℃ until complete evaporation of the solution.

The characterization methods and the preparation for the characterization is also not very well described. How did you prepare your samples for SEM and TEM? In which geometry did you measure XRD (powder diffraction in transmission)?

Due to the magnetic nature of the sample, we performed sample embedding treatment on the sample during SEM testing. The morphology and corresponding elemental analyses of samples after mounting were characterized by scanning electron microscopy (SEM) equipped with energy-dispersive X-ray spectroscopy (EDS; Quanta 250 FEG). Considering the magnetic properties of the sample, we used Lorentz-transmission electron microscopy for TEM, and there is no need for special treatment of the sample for this test. The X-ray diffraction (XRD) patterns were measured in the 2θ rang of 10-90° at a scan speed of 0.1°/s with Advance D8 using Cu Kα radiation. These have been made corresponding modifications in the manuscript and have been highlighted in red.

Which X-ray source has been used for XPS?

XPS analysis was characterized by an X-ray photoelectron spectrometry (XPS; ESCALAB250Xi), using Al Kα as x-ray source.

The same is true for IR spectroscopy. How did you conduct the measurements? In transmission geometry with KBr? Please state your preparation and experiments!

The chemical structure analyzed by Fourier transform infrared (FTIR) was collected on Bruker VERTEX 70 FTIR spectroscopy with a frequency in the range of 400 to 4000 cm-1, and a sample to KBr ratio of 1:100.

The same is true for Raman spectroscopy. At which power did you conduct your measurements (since they are prone to further oxidation)?

Raman studies with a 532 nm laser were measured from 100 to 1000 cm-1 using a Jobin Yvon France LABRAM Aramis Raman microscopy system.

For TGA and DSC it is not clear if the experiments have been conducted in inert gas or in air. Moreover, there is no statement for the heating rate.

The thermogravimetry and differential scanning calorimetry (TG-DSC) was performed by a thermal analysis instrument of NETZSCH STA 449C, in which the samples were heated from room temperature to 700 ℃ under an Ar atmosphere in alumina crucibles at a rate of 10 ℃/min.

Results and Discussion:

Can you discuss your XRD results with literature?

We discussed the results of XRD in the manuscript. And unexpectedly, only peaks originating from the raw powders were detected without the observation of any peaks corresponding to the CoFe2O4 phase of the oxidized powders. The reason for this result may be that the amount of CoFe2O4 phase is too low to be detected by XRD.

Figure 3: There is no scaling bar for the SEM image. Please add one.

We have added a scale bar in the SEM image.

In the EDS: Where exactly can one observe cobalt? I just see iron which strongly overlays cobalt.

    Due to the close energy positions of Fe and Co in the energy spectrum, it is difficult to distinguish them. I hope to receive your recognition, thank you very much.

Figure 4: The scaling bars are not readable. It is also not clear how you identified each element? Can you add how the color relates to each element? I think you related the coloring to a specific energy?

We have added a scale bar in the Figure. In the figure, Co, Fe, and O elements correspond to green, yellow, and blue, respectively.

Figure 5: You show particles larger than 1 µm. How can this be called a nanocomposite?

Based on your suggestions, we have modified the nonacomposites in the article to composites.

Figure 6: How did you fit your spectra? This is not described. Also it seems that one fit for Co2p3/2 is missing.

In XPS data fitting, we use the binding energy of C1s as the benchmark, which is 284.5 ev. We have added the location identification of Co2p3/2 in Figure 6 and discussed it in the manuscript accordingly.

Figure 7: Moreover, no peak corresponding to metallic Fe was observed for the sample oxidized by 2.5 wt% NH3·H2O, which suggests that a complete nano-CoFe2O4 layer was formed. How does this observation lead to this statement? You just observe iron. You observe oxides here and there might also be some carbon and therefore no elemental iron can be observed. Unless you show the cobalt oxide, you cannot make this statement.

With your suggestion, we have removed the sentence: Moreover, no peak corresponding to metallic Fe was observed for the sample oxidized by 2.5 wt% NH3·H2O, which suggests that a complete nano-CoFe2O4 layer was formed.

Figure 8: Can you please reference your peak assignments? “From the spectrum, the absorption bands at 1058, 779, and 465 cm-1 were assigned to Fe-O stretching” 1379 cm-1 is usually not assigned to an OH vibration. Also there is no Fe-O vibration at around 1000 cm-1. Both peaks rather indicate carbon, which is also visible in the XPS.

With your suggestion, we have made modifications to this section by consulting the references. The broad absorption bands were observed at 3437, 1637, 1379 and 1058 cm-1 due to the -OH, NH4+ and NH3 vibration, whose peaks are considered to be appeared due to H2O in the air and the adsorbed NH3 on the CoFe2O4/Fe composites [26, 27].

Figure 9: Can you turn the figure. From high to low wavenumbers?

Also can you please reference your spinel statement?

We have redrawn the figure 9, from high to low wavenumbers. We have added reference.

Figure 10: Please indicate the temperature in the figure caption.

Can you also compare your saturation magnetization as well as the remanence of your composites with literature?

    We have indicated the temperature in the figure caption. We have added the following content to the manuscript: The Ms values of the oxidized Fe powder are consistent with that of the raw powder (210 emu/g). This is attributed to the formation of the ferromagnetic CoFe2O4 coating by the in situ oxidation method for the prevention of magnetic dilution. As shown in the inset, the coercive field of the oxidized Fe powders increased with increasing ammonia concentration, mainly owing to the increasing content of CoFe2O4 in the insulating layer.

Figure 11: There are no units on the left y-axis. Please add the units to show the correct range.

Also it is unclear why the peak at 562 ℃ should lead to oxidation. One would expect an increase in weight if oxygen is taken up. Also what happens to the cobalt? Please reference your findings with observations in literature. Please also indicate the atmosphere which was used for your experiments.

We have redrawn the image and provided the corresponding data on the left y-axis. The DSC curve exhibits an endothermic peak at 562 ℃, which indicates that the CoFe2O4 began to decompose, leaving Fe2O3 [30]. The samples were heated from room temperature to 700 ℃ under an Ar atmosphere in alumina crucibles at a rate of 10 ℃/min.

No reference in XPS discussion.

We have added the references.

Figure 12: Legends and scaling bar are not readable. Can you increase the font for the x-axis of EDS and the font for the scaling bar (or mention the scale in the figure caption).

    We have made modifications to the Figure 12.

In addition, we have marked the revised parts of the article in red.

    We tried our best to improve the manuscript. Thank you very much for your comments and suggestions.  

Best wishes.

Sincerely

Shigeng Li, Fangping Ouyang

Professor of Materials Science

School of Physics and Electronics,

Central South University

Tel: +86-15079927360

Fax: +86-15079927360

Email: [email protected]; [email protected].

Round 2

Reviewer 1 Report

Modifications do not reply to the requested questions. The study of the nanostructures (interlayer) regards only to chemical composition and not the structural and magnetic properties. The role of the interlayers is indirect. This is not an article on nanomaterials.

Author Response

Dear Reviewer:

Thank you for your comments concerning our manuscript. Those comments are all valuable and very helpful for revising and improving our paper, as well as the important guiding significance to our researches.

We have made revisions to the manuscript based on your first round of review comments. The introduction has been supplemented and corresponding references have been added, which are consistent with the content of the manuscript. We were used micron level iron powder as the raw material, and used oxidation technology to form a nano thickness insulation layer on the surface. The insulation layer is a key factor in reducing the loss of magnetic material cores. Based on your and editor's suggestions, we have transferred the manuscript from journal Nanomaterials to journal Materials to make it more in line with the journal's thematic requirements.

Thank you very much for your comments and suggestions.  

Best wishes.

Sincerely

Shigeng Li, Fangping Ouyang

Professor of Materials Science

School of Physics and Electronics,

Central South University

Tel: +86-15079927360

Fax: +86-15079927360

Email: [email protected]; [email protected].

Reviewer 3 Report

has been improved

Author Response

Dear Reviewer:

Thank you for your comments concerning our manuscript. Those comments are all valuable and very helpful for revising and improving our paper, as well as the important guiding significance to our researches.

Thank you very much for your comments and suggestions.  

Best wishes.

Sincerely

Shigeng Li, Fangping Ouyang

Professor of Materials Science

School of Physics and Electronics,

Central South University

Tel: +86-15079927360

Fax: +86-15079927360

Email: [email protected]; [email protected].